# Diversity-Aware Recursive Feature Multiple Kernel Learning

**Nan Cao** [* 1]  **Xu Zhao** [* 1]  **Teng Zhang** [1]

## Abstract

*Multiple kernel learning* (MKL) combines several base kernels in the spirit of ensemble learning, yet existing methods rarely model kernel diversity—a known cornerstone of ensembles—and most traditional kernels weight all features uniformly, ignoring feature-level discriminability. We address both gaps with DARFMMKL: a data-driven kernel family (Recursive Feature Machine kernels) that learns feature importance directly from data, paired with a kernel selection method that jointly optimizes diversity and quality. The resulting NP-hard binary quadratic program is reformulated via Glover linearization and continuous relaxation into a linear program, and accelerated by Nyström sketching, yielding a selector whose cost is decoupled from the sample size. We provide a covering-number generalization bound that explicitly relates kernel diversity to estimation error. Experiments on 12 benchmark datasets show that DARFMMKL consistently outperforms 9 state-of-the-art MKL methods.

## 1. Introduction

Kernel methods (Schölkopf et al., 1999) have achieved significant success in various machine learning applications, including but not limited to classification, regression, ranking, clustering, and dimensionality reduction. The effectiveness of kernel methods is strongly influenced by the choice of kernel function, as the associated feature mapping has a profound impact on generalization performance. Consequently, the selection of an appropriate kernel plays an essential role in the success of kernel methods.

In the context of kernel learning, it is often more advantageous to select or combine multiple kernels rather than exhaustively searching for a single optimal kernel (Gönen & Alpaydin, 2011), giving rise to *Multiple kernel learning* (MKL). Most existing MKL methods aim to learn a weighted linear (Lanckriet et al., 2004; Rakotomamonjy et al., 2008) or non-linear combination (Varma & Babu, 2009; Cortes et al., 2009) of kernels from a predefined set by constructing an optimization problem. Some approaches assign kernel weights based on different measures of kernel quality, such as kernel alignment (Cristianini et al., 2001; Kandola et al., 2002; Qiu & Lane, 2009; Cortes et al., 2010; Alavi & Hashemi, 2022), feature space-based kernel matrix evaluation measures (Nguyen & Ho, 2008), and KL divergence (Ying et al., 2009), while others formulate different structural risk optimization problems, including semi-definite programming or quadratically constrained quadratic programming with various restrictions on kernel properties (Do et al., 2009; Kloft et al., 2011; Liu et al., 2017; Han et al., 2018). The primary limitation of these MKL methods lies in their insufficient consideration of diversity among kernel functions, susceptibility to disturbance from suboptimal kernels, and poor scalability to large datasets.

Numerous studies (Kuncheva & Whitaker, 2003) have demonstrated the importance of diversity among classifiers in ensemble learning (Zhou, 2012). Xia & Hoi (2012) and Shen et al. (2021) introduced ensemble methods into kernel learning, indirectly highlighting the importance of kernel diversity, but neither has explicitly quantified it. In the field of multiple kernel clustering, the Frobenius product and squared Frobenius divergence between kernel matrices have been used to represent kernel diversity (Liu et al., 2016; 2020; Yao et al., 2020). However, the computational cost remains expensive due to the alternating optimization procedure and calculations involving kernel matrices.

To address the computational burden arising from large datasets, several methods have been proposed, including random Fourier feature (RFF) methods (Rahimi & Recht, 2007; 2008), Nyström methods (Williams & Seeger, 2000; Drineas & Mahoney, 2005), and coreset methods (Tsang et al., 2005; 2006). The RFF method approximates shift-invariant kernels with orthogonal trigonometric function families and is data-independent, which results in weaker generalization performance. The coreset approximation method and the Nyström method differ essentially in their

---

[*]Equal contribution  [1]School of Computer Science and Technology, Huazhong University of Science and Technology, Wuhan, China. Correspondence to: Teng Zhang <tengzhang@hust.edu.cn>.

*Proceedings of the $43^{rd}$ International Conference on Machine Learning*, Seoul, South Korea. PMLR 306, 2026. Copyright 2026 by the author(s).

sample selection approaches, representing a trade-off between the quality of selected data points and computational cost.

In this paper, *Diversity-Aware Recursive Feature Machine Multiple Kernel Learning* (DARFMMKL) is proposed, which can either be used directly as an MKL method or serve as a preprocessing enhancement for other MKL methods. Specifically, a new RFM kernel is introduced to learn the key features from diverse features, and then both diversity and accuracy of the selected kernels are explicitly optimized over a large candidate set in an efficient manner. A disagreement measure between pairwise kernels is adopted as the diversity metric. Meanwhile, *Centered Kernel Alignment* (CKA) (Cortes et al., 2010) is employed as the accuracy measure, and these two components are jointly optimized. To alleviate the computational burden of MKL on large datasets, the sketching technique is introduced to reduce the size of the kernel matrix. The task is formalized as a *binary quadratic program* (BQP), which is generally NP-hard. Therefore, the problem is reformulated via Glover's method (Glover, 1975) and continuous relaxation into a *linear program* (LP), which can be solved efficiently.

Each individual ingredient—centered kernel alignment as a quality criterion (Cortes et al., 2010), the general MKL paradigm for combining base kernels (Gönen & Alpaydin, 2011), diversity-aware selection inspired by ensemble learning (Kuncheva & Whitaker, 2003), and Nyström sketching for kernel approximation (Williams & Seeger, 2000)—has been studied previously. The central technical contribution of this work is to integrate them into a single scalable MKL framework with a data-driven kernel family. Concretely, the contributions are as follows:

- A new RFM kernel family is introduced to MKL, in which the feature-importance matrix $\mathbf{W}$ is learned directly from data via the Average Gradient Outer Product (AGOP) mechanism (Radhakrishnan et al., 2024), in contrast to traditional kernels that treat all features uniformly.

- Kernel diversity is formally defined via pairwise prediction disagreement, and *joint* optimization of kernel quality (CKA) and kernel diversity is formulated as a binary quadratic program. To the best of our knowledge, this is the first MKL formulation that explicitly couples a diversity term with a CKA-based quality term inside a single selection objective.

- The resulting NP-hard BQP is reformulated via Glover linearization (Glover, 1975) and continuous relaxation into a linear program, and further accelerated by Nyström sketching, yielding a selector whose cost is decoupled from the sample size $n$.

- A generalization analysis based on covering numbers is provided that explicitly relates kernel diversity to the estimation error, and a complementary error decomposition (kernel selection / RFM feature learning / sketching) is given to make the role of each component transparent.

The remainder of this paper is organized as follows. First, some preliminaries are introduced. Then, the formulation and optimization of DARFMMKL are presented, followed by theoretical analysis of the generalization bound. Subsequently, experimental results and empirical observations are reported. Finally, the paper concludes with directions for future work.

## 2. Preliminaries

Throughout this paper, scalars are denoted by lowercase letters (e.g., $m$ and $M$), while vectors and matrices are denoted by boldface lowercase and uppercase letters, respectively (e.g., $\boldsymbol{x}$ and $\mathbf{X}$). In particular, $\mathbf{1}$ denotes the all-ones vector. Sets are designated by uppercase letters in calligraphic font (e.g., $\mathcal{S}$), and the integer set $\{1, \ldots, M\}$ is denoted by $[M]$. The input space is $\mathcal{X} \subseteq \mathbb{R}^d$, and $\mathcal{Y} = \{1, -1\}$ denotes the label set. Let $\mathcal{D}$ be an unknown (underlying) distribution over $\mathcal{X} \times \mathcal{Y}$. All data are assumed to be drawn independently and identically distributed (i.i.d.) according to $\mathcal{D}$. For the feature mapping $\phi : \mathcal{X} \mapsto \mathbb{H}$ associated with a positive definite kernel $\kappa$, where $\mathbb{H}$ is the corresponding reproducing kernel Hilbert space (RKHS), the relation $\kappa(\boldsymbol{x}_i, \boldsymbol{x}_j) = \langle \phi(\boldsymbol{x}_i), \phi(\boldsymbol{x}_j) \rangle_{\mathbb{H}}$ holds for any $\boldsymbol{x}_i$ and $\boldsymbol{x}_j$. Given a dataset $\mathcal{S}$ with $n$ elements, the corresponding kernel matrix $\mathbf{K}$ has its $(i, j)$-th entry defined as $\kappa(\boldsymbol{x}_i, \boldsymbol{x}_j)$. The indicator function $\mathbb{I}(\cdot)$ returns 1 if the argument is true and 0 otherwise.

### 2.1. Kernel Alignment

Given a dataset $\mathcal{S}$ with $n$ elements, for any two kernels $\kappa_i$ and $\kappa_j$, the kernel alignment $A : \mathbb{R}^{n \times n} \times \mathbb{R}^{n \times n} \mapsto \mathbb{R}$ measures their similarity via the corresponding kernel matrices $\mathbf{K}_i$ and $\mathbf{K}_j$ as follows (Cristianini et al., 2001):

$$A(\mathbf{K}_i, \mathbf{K}_j) = \frac{\langle \mathbf{K}_i, \mathbf{K}_j \rangle_F}{\sqrt{\langle \mathbf{K}_i, \mathbf{K}_i \rangle_F \langle \mathbf{K}_j, \mathbf{K}_j \rangle_F}},$$

where $\langle \cdot, \cdot \rangle_F$ denotes the Frobenius product between matrices. If $\mathbf{K}_j$ is chosen as the ground-truth label matrix $\boldsymbol{y}\boldsymbol{y}^\top$, then $A(\mathbf{K}_i, \boldsymbol{y}\boldsymbol{y}^\top)$ can be interpreted as a quality measure of $\kappa_i$.

The kernel alignment suffers from the ill-conditioned matrix problem when the input kernel matrices are nearly singular. To address this issue, the centered kernel alignment was

proposed (Cortes et al., 2010), which is defined as

$$A^c(\mathbf{K}_i, \mathbf{K}_j) = \frac{\langle \mathbf{K}_i^c, \mathbf{K}_j^c \rangle_F}{\sqrt{\langle \mathbf{K}_i^c, \mathbf{K}_i^c \rangle_F \langle \mathbf{K}_j^c, \mathbf{K}_j^c \rangle_F}},$$

where $\mathbf{K}_\ell^c = [\mathbf{I}_n - \mathbf{1}\mathbf{1}^\top/n]\mathbf{K}_\ell[\mathbf{I}_n - \mathbf{1}\mathbf{1}^\top/n]$ for $\ell \in \{i, j\}$. The centered kernel alignment has been shown to exhibit improved generalization performance.

## 3. Formulation

Given a set of kernels $\mathcal{M} = \{\kappa_1, \ldots, \kappa_M\}$, the goal is to select exactly $m$ high-quality kernels that exhibit diversity. To begin, we define the difference between two kernels $\kappa_i$ and $\kappa_j$ on a validation dataset $\mathcal{V}$ as

$$\text{Diff}(\kappa_i, \kappa_j) = \frac{1}{|\mathcal{V}|} \sum_{\boldsymbol{x} \in \mathcal{V}} \mathbb{I}(h_i(\boldsymbol{x}) \neq h_j(\boldsymbol{x})),$$

where $h_i$ and $h_j$ are the learned classifiers in $\mathbb{H}_i$ and $\mathbb{H}_j$, respectively. By denoting the selected kernel set as $\mathcal{K}$ and $\eta_i = \mathbb{I}(\kappa_i \in \mathcal{K})$ where $\sum_{i \in [M]} \eta_i = m$, the diversity of $\mathcal{K}$ is defined as

$$\text{Div}(\mathcal{K}) = \sum_{i \in [M]} \sum_{j \in [M]} \eta_i \eta_j \text{Diff}(\kappa_i, \kappa_j).$$

This ordered-pair convention counts each unordered pair twice; it only rescales the diversity term and is absorbed by the trade-off parameter $\lambda$. In the theoretical bound below, we use the corresponding unordered-pair quantity $\text{Div}_u(\mathcal{K}) = \sum_{i<j} \eta_i \eta_j \text{Diff}(\kappa_i, \kappa_j)$ so that the number of selected pairs is $m_c = \binom{m}{2}$.

To ensure high quality of the selected kernels, the alignment between them and the ground-truth label matrix $\boldsymbol{y}\boldsymbol{y}^\top$ is maximized. Therefore, the goal can be formulated as

$$\max_{\boldsymbol{\eta}} \text{Div}(\mathcal{K}) + \lambda \sum_{i \in [M]} \eta_i A^c(\mathbf{K}_i, \boldsymbol{y}\boldsymbol{y}^\top)$$

$$\text{s.t.} \sum_{i \in [M]} \eta_i = m, \ \eta_i \in \{0, 1\}, \ i \in [M], \qquad (1)$$

where $\lambda$ is a trade-off parameter that balances diversity and quality.

By denoting $G_{ij} = \text{Diff}(\kappa_i, \kappa_j)$ and $b_i = \lambda A^c(\mathbf{K}_i, \boldsymbol{y}\boldsymbol{y}^\top)$, Eqn. (1) can be rewritten as

$$\max_{\boldsymbol{\eta}} f(\boldsymbol{\eta}) = \sum_{j \in [M]} \sum_{i \in [M]} G_{ij} \eta_i \eta_j + \sum_{j \in [M]} b_j \eta_j \qquad (2)$$

$$\text{s.t.} \ \mathbf{1}^\top \boldsymbol{\eta} = m, \boldsymbol{\eta} \in \{0, 1\}^M$$

Note that constructing both $\mathbf{G}$ and $\boldsymbol{b}$ requires computing the kernel matrix $\mathbf{K}$, which can be expensive. Therefore, the sketching method, a widely used low-rank matrix approximation technique, is employed for improved computational efficiency. For any square matrix $\mathbf{K} \in \mathbb{R}^{n \times n}$, the approximated matrix after sketching is $\tilde{\mathbf{K}} = \mathbf{R}\mathbf{K}\mathbf{R}^\top \in \mathbb{R}^{s \times s}$, where $\mathbf{R} \in \mathbb{R}^{s \times n}$ is the sketching matrix and $s \ll n$. There are three commonly used approaches to obtain the sketching matrix: sub-Gaussian sketch, randomized orthogonal system sketch, and Nyström sketch. In this paper, the Nyström sketch is adopted due to its simplicity.

### 3.1. Recursive Feature Machine Kernels

Conventional kernel methods, such as those based on RBF and polynomial kernels, typically rely on predefined kernel functions with manually specified hyperparameters, thereby treating all input features as equally important. However, in real-world tasks, some features can be highly discriminative for the prediction task while others are irrelevant or even noisy. To address this fundamental limitation, a new class of data-driven kernels is introduced based on the Recursive Feature Machine (RFM) framework (Radhakrishnan et al., 2024), which enables feature importance to be learned directly from data.

The key insight underlying this approach is to replace the standard Euclidean distance employed in classical kernels with a learned Mahalanobis distance that captures feature importance. Given a positive semi-definite matrix $\mathbf{W} \in \mathbb{R}^{d \times d}$, the *RFM-Gaussian kernel* and the *RFM-Laplacian kernel* are defined as:

$$\kappa_{\mathbf{W}, \sigma}(\boldsymbol{x}_i, \boldsymbol{x}_j) = \exp\left(-\frac{(\boldsymbol{x}_i - \boldsymbol{x}_j)^\top \mathbf{W}(\boldsymbol{x}_i - \boldsymbol{x}_j)}{2\sigma^2}\right) \quad (3)$$

$$\kappa_{\mathbf{W}, \sigma}^L(\boldsymbol{x}_i, \boldsymbol{x}_j) = \exp\left(-\frac{\|\mathbf{W}^{1/2}(\boldsymbol{x}_i - \boldsymbol{x}_j)\|_2}{\sigma}\right) \quad (4)$$

where $\sigma = \beta d_m$ denotes the bandwidth parameter. $\beta$ is the scaling factor, and $d_m$ is the median of the Mahalanobis distances. $\mathbf{W}^{1/2}$ denotes the principal square root of $\mathbf{W}$. These kernels are guaranteed to be positive semi-definite, as they can be interpreted as standard Gaussian or Laplacian kernels applied to the transformed features $\tilde{\boldsymbol{x}} = \mathbf{W}^{1/2}\boldsymbol{x}$. The matrix square root is computed via eigenvalue decomposition: $\mathbf{W}^{1/2} = \mathbf{V} \text{diag}(\sqrt{\boldsymbol{\lambda}})\mathbf{V}^\top$, where $\mathbf{V}$ and $\boldsymbol{\lambda}$ denote the eigenvectors and eigenvalues of $\mathbf{W}$, respectively (Radhakrishnan et al., 2024).

To instantiate the kernel, we still need to specify the feature importance matrix $\mathbf{W}$. The RFM framework addresses this through the Average Gradient Outer Product (AGOP) mechanism. Given a predictor $f : \mathbb{R}^d \to \mathbb{R}^c$ that maps $d$-dimensional features to $c$ class outputs, the Jacobian matrix $\mathbf{J}(\boldsymbol{x}) \in \mathbb{R}^{c \times d}$ captures the sensitivity of each output to

perturbations in each input feature:

$$\mathbf{J}(\boldsymbol{x})_{r,l} = \frac{\partial f_r(\boldsymbol{x})}{\partial x_l}, \quad r \in [c], l \in [d] \tag{5}$$

The AGOP is then defined as the expected outer product of the Jacobian over the data distribution:

$$\mathbf{P} = \mathbb{E}_{\boldsymbol{x}}[\mathbf{J}(\boldsymbol{x})^\top \mathbf{J}(\boldsymbol{x})] \approx \frac{1}{n} \sum_{i=1}^{n} \mathbf{J}(\boldsymbol{x}_i)^\top \mathbf{J}(\boldsymbol{x}_i) \tag{6}$$

The resulting matrix $\mathbf{P} \in \mathbb{R}^{d \times d}$ is guaranteed to be positive semi-definite. The diagonal entry $\mathbf{P}_{ll}$ measures the cumulative squared sensitivity of all outputs to the $l$-th feature, thereby indicating which feature dimensions are most discriminative. The off-diagonal entries $\mathbf{P}_{ll'}$ capture feature co-importance, reflecting correlated feature directions that jointly influence predictions.

For kernel classifiers of the form

$$f(\boldsymbol{x}) = \sum_{i=1}^{n} \alpha_i y_i \kappa(\boldsymbol{x}_i, \boldsymbol{x}),$$

the AGOP can be computed analytically by differentiating the kernel function with respect to the input features (Radhakrishnan et al., 2024). This analytical formulation enables efficient feature importance learning without requiring gradient backpropagation.

### 3.1.1. ITERATIVE RFM LEARNING

In the RFM framework, the matrix $\mathbf{W}$ is learned through an iterative procedure. Beginning with $\mathbf{W}_0 = \mathbf{I}$ (corresponding to the standard Euclidean distance), each iteration comprises three steps: we first train a kernel classifier using the current Mahalanobis kernel $\kappa_{\mathbf{W}_t, \sigma}$. Then the matrix $\mathbf{P}_t$ is calculated from the trained classifier using Eqn. (6). Finally, we update the feature importance matrix as $\mathbf{W}_{t+1} = \mathbf{P}_t$.

After $T$ iterations, the obtained matrix $\mathbf{W}_T$ captures the task-relevant feature structure learned from the data. A recent study published in *Science* (Radhakrishnan et al., 2024) demonstrated that this AGOP mechanism characterizes feature learning across diverse neural network architectures, including Transformers, CNNs, and MLPs.

To make the iterative procedure mathematically well-posed, we further show that, under mild assumptions, the normalized RFM update admits at least one fixed point. Let $\mathcal{C} = \{\mathbf{W} \in \mathbb{R}^{d \times d} : \mathbf{W} \succeq 0, \max_l \mathbf{W}_{ll} \leq 1\}$ be the compact convex set of normalized PSD matrices, and define the per-iteration map $\Phi(\mathbf{W}) = \mathbf{P}(\mathbf{W})/(\max_l \mathbf{P}_{ll}(\mathbf{W}) + \epsilon)$, where $\mathbf{P}(\mathbf{W})$ is the AGOP of the kernel ridge classifier trained with kernel $\kappa_{\mathbf{W}, \sigma}$ and $\epsilon > 0$ is a small constant. The normalization only rescales $\mathbf{W}$ by a positive scalar; since the bandwidth $\sigma = \beta d_m$ adapts to the median Mahalanobis distance, this scaling is absorbed by $\sigma$ and leaves

the resulting kernel $\kappa_{\mathbf{W}, \sigma}$ unchanged. Hence analyzing the normalized map $\Phi$ is equivalent to analyzing the update $\mathbf{W}_{t+1} = \mathbf{P}_t$ used in practice, while additionally ensuring boundedness.

**Proposition 3.1** (Boundedness and Fixed-Point Existence). *Assume* (A1) *the training data are bounded,* $\|\boldsymbol{x}_i\| \leq R$ *for all* $i \in [n]$; (A2) *the bandwidth rule* $\sigma(\mathbf{W})$ *is continuous on* $\mathcal{C}$ *and bounded away from zero,* $\sigma(\mathbf{W}) \geq \sigma_{\min} > 0$ *(in our implementation,* $\sigma$ *is computed from the median Mahalanobis distance and is clipped from below by a small constant* $\sigma_{\min}$, *so this assumption holds by construction);* (A3) *the kernel ridge regularizer satisfies* $\lambda_{\mathrm{rfm}} > 0$; *and* (A4) *the RFM kernel used in the update is continuously differentiable in its input and in* $\mathbf{W}$ *on* $\mathcal{C}$ *(this holds directly for the Gaussian-RFM kernel, while a smoothed Laplacian variant satisfies the same condition). Then* (i) $\mathbf{P}(\mathbf{W}) \succeq 0$ *for all* $\mathbf{W} \in \mathcal{C}$; (ii) $\Phi$ *maps* $\mathcal{C}$ *into itself,* $\Phi(\mathcal{C}) \subseteq \mathcal{C}$; (iii) $\Phi : \mathcal{C} \to \mathcal{C}$ *is continuous; and* (iv) *by Brouwer's fixed-point theorem,* $\Phi$ *admits at least one fixed point* $\mathbf{W}^\star \in \mathcal{C}$.

*Proof Sketch.* $\mathcal{C}$ is compact and convex (intersection of the PSD cone with $\{\mathbf{W}_{ll} \leq 1\}$). $\mathbf{P}(\mathbf{W})$ is PSD as a nonnegative average of $\mathbf{J}(\boldsymbol{x}_i; \mathbf{W})^\top \mathbf{J}(\boldsymbol{x}_i; \mathbf{W})$, so $\Phi(\mathbf{W}) \succeq 0$ and $\max_l \Phi(\mathbf{W})_{ll} < 1$, giving $\Phi(\mathcal{C}) \subseteq \mathcal{C}$. Continuity follows by composing the continuous chain $\mathbf{W} \mapsto \mathbf{K}_{\mathbf{W}} \mapsto (\mathbf{K}_{\mathbf{W}} + \lambda_{\mathrm{rfm}} \mathbf{I})^{-1} \mapsto \boldsymbol{\alpha}(\mathbf{W}) \mapsto \mathbf{J}(\mathbf{W}) \mapsto \mathbf{P}(\mathbf{W}) \mapsto \Phi(\mathbf{W})$, using (A1)–(A4). Brouwer's theorem (Brouwer, 1911) then yields a fixed point $\mathbf{W}^\star \in \mathcal{C}$. The complete argument is given in Appendix A. $\square$

This result applies directly to differentiable RFM kernel instantiations and guarantees that the normalized update is well-posed and bounded. Empirical convergence is rapid: the Frobenius distance $\|\mathbf{W}_T - \mathbf{W}_{T-1}\|_F$ drops from $\sim 10$ at $T{=}1$ to $\sim 10^{-5}$ by $T{=}3$ on representative datasets (Appendix C).

### 3.1.2. INTEGRATION WITH MULTIPLE KERNEL LEARNING

The RFM kernels provide a fundamentally different source of diversity compared to traditional kernels. Whereas RBF kernels with varying bandwidth $\sigma$ differ only in their smoothness, and polynomial kernels with varying degrees differ only in their nonlinearity, RFM kernels with different learned $\mathbf{W}$ matrices differ in their *feature weighting*. This learned weighting is adapted to the specific discriminative structure of each dataset.

The candidate kernel pool $\mathcal{M}$ is enriched by including RFM kernels alongside traditional kernels. Specifically, for each combination of kernel type (Gaussian or Laplacian), bandwidth $\sigma$, and RFM iteration $t \in \{1, \dots, T\}$, a distinct RFM kernel is generated. Compared with the traditional kernels, the RFM kernels capture the discriminative features, and

show greater adaptation to specific tasks since they learn feature representations rather than mere parameter tuning. This integration preserves the efficiency of the sketching-based kernel selection framework while substantially expanding the expressiveness of the selected kernel combination.

## 4. Optimization

Note that $\mathbf{G}$ is a symmetric matrix with $G_{ij} \in [0,1]$ and $G_{ii} = 0$, so $\mathbf{G}$ is generally indefinite (a non-zero symmetric matrix with zero diagonal must have both positive and negative eigenvalues). It is evident that Eqn. (2) is a non-convex BQP problem, which is generally NP-hard. To solve it efficiently, the problem is reformulated using Glover linearization (Glover, 1975) and relaxed into a continuous optimization.

Specifically, each $\eta_j(\sum_{i \in [M]} G_{ij} \eta_i)$ in Eqn. (2) is initially replaced with a continuous variable $z_j$, and four linear constraints are used to bound the gap. Thus, Eqn. (2) can be reformulated as the following mixed integer linear programming (MILP) problem:

$$\max_{\boldsymbol{\eta}, \boldsymbol{z}} f_1(\boldsymbol{\eta}, \boldsymbol{z}) = \sum_{j \in [M]} z_j + \sum_{j \in [M]} b_j \eta_j \tag{7}$$

$$\text{s.t. } z_j \leq U_j^1 \eta_j, \ z_j \leq \sum_{i \in [M]} G_{ij} \eta_i - L_j^0(1 - \eta_j) \tag{8}$$

$$z_j \geq L_j^1 \eta_j, \ z_j \geq \sum_{i \in [M]} G_{ij} \eta_i - U_j^0(1 - \eta_j) \tag{9}$$

$$\sum_{i \in [M]} \eta_i = m, \ \eta_i \in \{0, 1\} \tag{10}$$

For each $j$, $U_j^0$, $U_j^1$, $L_j^0$, and $L_j^1$ are the upper and lower bounds on $\sum_{i \in [M]} G_{ij} \eta_i$ when $\eta_j$ is 0 and 1, respectively. Eqn. (7) with an additional $M$ unrestricted auxiliary continuous variables and $4M$ auxiliary constraints is equivalent to Eqn. (2) if constraints (8)–(10) can guarantee that $z_j = \eta_j(\sum_{i \in [M]} G_{ij} \eta_i), \forall j$. The upper and lower bounds for each $j$, denoted as $U_j^0$, $U_j^1$, $L_j^0$, and $L_j^1$ in Eqn. (7), can be calculated using various methods (Adams et al., 2004). However, for computational efficiency, these bounds are calculated using the method originally proposed by Glover (1975), as $L_j^1 = L_j^0 = \sum_{i \in [M], G_{ij} < 0} G_{ij}$ and $U_j^1 = U_j^0 = \sum_{i \in [M], G_{ij} > 0} G_{ij}$.

As indicated by Adams & Forrester (2005), the constraints in Eqn. (9), which restrict the lower bound of $z_j$, can be removed to reduce the computational complexity of problem (7) due to the maximization objective. Moreover, by substituting the variable $z_j$ with $s_j = U_j^1 \eta_j - z_j$, the number of constraints can be further reduced. The problem can

be rewritten as

$$\max_{\boldsymbol{\eta}, \boldsymbol{s} \geq 0} f_2(\boldsymbol{\eta}, \boldsymbol{s}) = \sum_{j \in [M]} (U_j^1 + b_j) \eta_j - \sum_{j \in [M]} s_j$$

$$\text{s.t. } s_j \geq (U_j^1 - L_j^0) \eta_j - \sum_{i \in [M]} G_{ij} \eta_i + L_j^0 \tag{11}$$

$$\sum_{i \in [M]} \eta_i = m, \eta_i \in \{0, 1\}$$

It should be noted that all elements of matrix $\mathbf{G}$ are within the range of $[0,1]$, so $L_j^1 = L_j^0 = 0$. After continuous relaxation of the constraints to $\boldsymbol{\eta} \in [0,1]^M$, the problem to be solved becomes

$$\max_{\boldsymbol{\eta}, \boldsymbol{s} \geq 0} f_3(\boldsymbol{\eta}, \boldsymbol{s}) = \sum_{j \in [M]} (U_j^1 + b_j) \eta_j - \sum_{j \in [M]} s_j \tag{12}$$

$$\text{s.t. } s_j \geq U_j^1 \eta_j - \sum_{i \in [M]} G_{ij} \eta_i, \ \sum_{i \in [M]} \eta_i = m, \eta_i \in [0, 1] \tag{13}$$

The top $m$ values in $\boldsymbol{\eta}$ are then selected to obtain the $m$ high-quality kernels. Let $\mathcal{I}^\star \subseteq [M]$ denote the index set of the selected kernels with $|\mathcal{I}^\star| = m$. Unless otherwise stated, the final composite kernel used by the downstream learner is the uniform average of the selected base kernels,

$$\kappa^\star(\boldsymbol{x}, \boldsymbol{x}') = \frac{1}{m} \sum_{i \in \mathcal{I}^\star} \kappa_i(\boldsymbol{x}, \boldsymbol{x}'), \tag{14}$$

which is positive semi-definite as a non-negative combination of PSD kernels. This uniform-averaging choice keeps the selection step decoupled from the learner and ensures that the reported gains can be attributed to which kernels are picked, rather than to a separate weight-tuning stage; the framework, however, is fully compatible with CKA-weighted or learner-weighted combinations and can be used as a preprocessing module for any MKL backend.

The MILP reformulations in Eqn. (7) and Eqn. (11) are tighter than the LP relaxation in Eqn. (12), while the latter can be solved more quickly. Although current optimization solvers are capable of efficiently solving BQP problems[1], Forrester & Hunt-Isaak (2020) indicate that directly presenting the original problem to the optimizer does not result in as stable and rapid solving as providing a reformulated linear representation of the problem. The entire kernel selection process is outlined in Algorithm 1.

## 5. Theoretical Analysis

In this section, generalization error bounds are derived in the standard agnostic learning setting. The aim is to

---

[1]https://mattmilten.github.io/mittelmann-plots/

**Algorithm 1** Unified Randomized Sketches Kernel Selection

---
**Input**: Set of base kernel functions $\mathcal{M}$, number of selected kernels $m$, trade-off parameter $\lambda$, sketch size $s$
**Output**: Combined kernel functions

1: Randomly sample $s$ data points from $\mathcal{V}$ according to the sampling matrix $\mathbf{Q}$ and construct the $\mathbf{S} \in \mathbb{R}^{s \times s}$.
2: Compute the approximate ideal kernel matrix $\tilde{\mathbf{K}}_{\boldsymbol{y}}$ and kernel matrix $\tilde{\mathbf{K}}$ with the $s$ sampling data points.
3: Compute the CKA between $\tilde{\mathbf{K}}$ and $\tilde{\mathbf{K}}_{\boldsymbol{y}}$ to obtain $\boldsymbol{b}$.
4: Train the classifier with $\tilde{\mathbf{K}}$ and compute the diversity of kernels $\mathbf{G}$.
5: Solve the LP and calculate the final combined kernel function.

---

bound the estimation error, which is the difference between the expected error $R(h)$ and the empirical margin error $\hat{R}^{\gamma}(h)$, where $R(h) = \Pr_{\mathcal{D}}(yh(\boldsymbol{x}) \leq 0)$ and $\hat{R}^{\gamma}(h) = |\{i \mid y_i h(\boldsymbol{x}_i) < \gamma\}|/n$. The selected kernel set is considered as a family $\mathcal{K} \subseteq \{\kappa : \mathcal{X} \times \mathcal{X} \to \mathbb{R}\}$, and the corresponding predictor class is defined as $\mathcal{F}_{\mathcal{K}} = \cup_{\kappa \in \mathcal{K}} \mathcal{F}_{\kappa}$. The main challenge in deriving such bounds is bounding the estimation error uniformly over all predictors in a class and exploring how the diversity considered in this method can enhance generalization performance. This is approached by using *covering numbers* (Vapnik, 1998). Specifically, we study coverings of classes of predictors and kernel families under the metric defined as

$$d_{\infty}^{\mathcal{S}}(h, \tilde{h}) = \max_i |h(\boldsymbol{x}_i) - \tilde{h}(\boldsymbol{x}_i)|, \; D_{\infty}^{\mathcal{S}}(\kappa, \tilde{\kappa})$$
$$= \max_{i,j} |\kappa(\boldsymbol{x}_i, \boldsymbol{x}_j) - \tilde{\kappa}(\boldsymbol{x}_i, \boldsymbol{x}_j)|$$

**Definition 5.1.** The uniform $l_{\infty}$ covering number $\mathcal{N}_n(\mathcal{F}, \epsilon)$ of a predictor class $\mathcal{F}$ considering all sample sets of size $n$: $\mathcal{N}_n(\mathcal{F}, \epsilon) = \sup_{|\mathcal{S}|=n} \mathcal{N}_{d_{\infty}^{\mathcal{S}}}(\mathcal{F}, \epsilon)$

**Definition 5.2.** The uniform $l_{\infty}$ kernel covering number $\mathcal{N}_n^D(\mathcal{K}, \epsilon)$ of a kernel class $\mathcal{K}$ considering all sample sets of size $n$: $\mathcal{N}_n^D(\mathcal{K}, \epsilon) = \sup_{|\mathcal{S}|=n} \mathcal{N}_{D_{\infty}^{\mathcal{S}}}(\mathcal{K}, \epsilon)$

**Definition 5.3.** The kernel class $\mathcal{K}$ pseudo-shatters a set of $n$ pairs of points $(\boldsymbol{x}_1, \boldsymbol{x}_1'), \ldots, (\boldsymbol{x}_n, \boldsymbol{x}_n')$ if there exist thresholds $t_1, \ldots, t_n \in \mathbb{R}$ such that for any $\varepsilon_1, \ldots, \varepsilon_n \in \{\pm 1\}$ there exists $\kappa \in \mathcal{K}$ with $\text{sign}(\kappa(\boldsymbol{x}_i, \boldsymbol{x}_i') - t_i) = \varepsilon_i$. The pseudodimension $d_{\phi}(\mathcal{K})$ is the largest $n$ such that there exists a set of $n$ pairs of points that are pseudo-shattered by $\mathcal{K}$.

**Theorem 5.4.** *For a family of kernels $\mathcal{K}$ bounded by $B$ and with a fixed pseudodimension $d_{\phi}$, suppose $\text{Div}_{\mathrm{u}}(\mathcal{K}) \geq q$ and the logarithm arguments below are greater than one. For any $\gamma > 0$, with probability at least $1 - \delta$ over $\mathcal{S}$, the following bound holds uniformly for all $h \in \mathcal{F}_{\mathcal{K}}$:*

$$R(h) - \hat{R}^{\gamma}(h) \leq \sqrt{8 \frac{2 + Q_1 + Q_2 - \log \delta}{n}} \quad (15)$$

*where*

$$Q_1 = d_{\phi} \log \frac{64 e m_c n^3 B}{q d_{\phi}},$$
$$Q_2 = \frac{64 m_c^2 B}{q^2} \log \frac{q e n}{4 m_c \sqrt{B}} \log \frac{128 m_c^2 n B}{q^2},$$

*and $m_c = \binom{m}{2}$.*

*Proof Sketch.* To begin, the generalization error of the predictor class is bounded based on its covering number. Next, the predictor class covering numbers are connected to kernel set diversity. Then, the pseudo-dimension is linked to kernel class covering numbers. Finally, combining all these elements completes the proof. $\qquad \square$

The bound shows how the diversity level enters the denominator-dominated terms of the covering-number analysis and can tighten the estimate when the logarithmic factors remain moderate. This bound can be translated to those presented in previous work (Lanckriet et al., 2004; Srebro & Ben-David, 2006) with slight modifications to the learning setting.

**Error decomposition.** Theorem 5.4 characterizes the *kernel-selection* step, while the RFM feature-learning and Nyström sketching components are addressed separately: Proposition 3.1 guarantees well-posedness of the RFM update (with empirical recovery shown in Appendix G), and the standard Nyström approximation bound (Drineas & Mahoney, 2005) controls the sketching perturbation. Because the selector relies on the *relative* ranking of kernels rather than exact values, moderate sketch sizes suffice in practice even when the absolute approximation error is nonnegligible. A complete component-wise error decomposition is given in Appendix B.

## 6. Experiments

In this section, the proposed method is experimentally evaluated on real-world datasets.[2]

### 6.1. Settings

#### 6.1.1. DATASETS

Twelve benchmark LIBSVM datasets used in the experiments are shown in Table 1. The dataset sizes range from 128 to 49,749, and the dimensionality ranges from 6 to 300, covering a broad range of properties. All features are normalized to the interval $[0, 1]$.

---
[2]Our implementation is available at https://github.com/LittleSpecial/DARFMMKL.

| Datasets | #Instance | #Feature |
|---|---|---|
| mux6-ver1 | 128 | 6 |
| sonar | 208 | 60 |
| heart | 270 | 13 |
| semeion | 319 | 256 |
| ionosphere | 351 | 34 |
| breast-cancer | 683 | 10 |
| australian | 690 | 14 |
| diabetes | 768 | 8 |
| german.numer | 1000 | 24 |
| splice | 3175 | 60 |
| a8a | 32561 | 123 |
| w7a | 49749 | 300 |

*Table 1.* Summary of datasets

### 6.1.2. BASELINES

The proposed DARFMMKL is compared with 5 one-step MKL methods and 4 two-step MKL methods: RBMKL: A SVM is trained with the mean of the combined kernels. PWMKL (Tanabe et al., 2008): Weights are assigned proportionally to kernel performance. ABMKL (Qiu & Lane, 2009): Weights are assigned proportionally to the kernel target alignment with the ideal kernel. EAMKL (Aiolli & Donini, 2015): The combination of kernels that maximizes the margin between classes is found. BFMKL (Alavi & Hashemi, 2022): Data relationships are captured via global and local kernel alignments, and the reliability of training samples is evaluated via self-paced learning. SimpleMKL (Rakotomamonjy et al., 2008): The combination of kernels is optimized using the reduced gradient. RMKL (Do et al., 2009): The combination of kernels that maximizes the margin between classes is found. GRMKL (Lauriola et al., 2017): The combination of kernels that maximizes the margin between classes while minimizing the radius is found. MEMKL (Lauriola et al., 2018): The combination of kernels that maximizes the margin between classes while minimizing the empirical complexity is found. The former five methods are one-step methods, and the latter four are two-step methods.

### 6.1.3. HYPERPARAMETERS

We tune $M \in \{100, 200, 300, 400\}$, $m \in \{10, 20, \ldots, 100\}$, and $\lambda \in \{M/16, M/8, M/4, M/2\}$. The candidate pool consists of $M$ Gaussian kernels $K(\boldsymbol{x}, \boldsymbol{x}') = \exp(-\tau \|\boldsymbol{x} - \boldsymbol{x}'\|_2^2)$ and polynomial kernels $K(\boldsymbol{x}, \boldsymbol{x}') = (1 + \boldsymbol{x}^\top \boldsymbol{x}')^d$, with $\tau$ taken from a geometric progression in $[2^{-8}, 2^8]$ and $d$ from a linear progression in $[1, 20]$. For the RFM kernels, we additionally include $M$ Gaussian-RFM (Eqn. 3) and Laplacian-RFM (Eqn. 4) kernels with $\mathbf{W}$ learned via kernel ridge regression and

the scaling factor $\beta$ tuned from $\{0.25, 0.5, \ldots, 10\}$. We use Nyström sketching with size $s \in \{10, 20, \ldots, 100\}$. SVM is used as the base learner for all methods except EAMKL, which uses kernel optimization of the margin distribution (KOMD) following its original paper. The regularization parameter $C$ for all methods is tuned from $\{1, 10, 100, 1000\}$. For convenience, the maximum iteration count and learning rate for GRMKL, RMKL, and MEMKL are set to 1000 and 0.1, respectively. The six parameters $\rho_1$, $\rho_2$, $\lambda$, $k$, $\gamma$, and $\eta$ for BFMKL are consistent with those in the original paper. Except for SimpleMKL and BFMKL, other comparison methods are implemented based on MKLpy (Lauriola & Aiolli, 2020). Each experiment is conducted using 3-fold cross-validation and repeated 10 times. All statements regarding statistical significance refer to a 95% level of significance. All experiments are run on a server equipped with an Intel(R) Xeon(R) Gold 5117 CPU @ 2.00GHz with 28 physical cores.

### 6.2. Results

The test accuracies of all compared methods on the 12 datasets are shown in Table 2. The time costs of all methods on the 8 regular-sized datasets when $M = 400$ are shown in Figure 1. Several observations can be made: In terms of accuracy, two-step MKL methods generally demonstrate superior performance compared to one-step MKL methods on regular-sized datasets, with the exception of SimpleMKL due to its use of the reduced gradient. The proposed method outperforms state-of-the-art MKL methods. In terms of time cost, BFMKL and the four two-step methods fail to produce results in time due to their iterative computation of numerous kernel matrices. In contrast, DARFMMKL exhibits consistent runtime that scales with $M$ and $s$ rather than the dataset size $n$, as confirmed on the large-scale datasets.

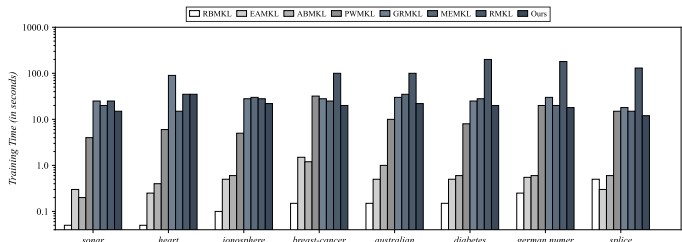

*Figure 1.* Training time of the compared methods and DARFMMKL on 8 datasets.

### 6.3. The Impact of Parameters

We study the parameter $m$ and the hyperparameter $\lambda$ on the 7 datasets (Figure 2 and Figure 3(a); $C = 1000$, $M = 400$, $s = 90$, with $\lambda = M/4$ and $m = 10$ fixed in turn).

Accuracy varies only mildly with $m$ and $\lambda$, indicating robustness; performance degrades once $m$ exceeds the optimal

| Datasets | RBMKL | EAMKL | PWMKL | ABMKL | BFMKL |
|---|---|---|---|---|---|
| mux6-ver1 | 87.51±3.40● | 88.35±4.34● | 86.18±4.28● | 88.72±3.98● | 82.72±4.39● |
| sonar | 84.48±4.40● | 79.22±4.79● | 84.14±4.49● | 83.75±4.31● | 84.71±4.29● |
| heart | 82.56±3.19● | 81.41±3.71● | 82.85±3.66● | 81.48±3.13● | 82.11±2.92● |
| semeion | 94.42±2.29● | 93.48±2.37● | 94.82±3.01● | 94.43±2.28● | 94.38±2.27● |
| ionosphere | 94.61±1.60● | 94.40±1.28● | 94.70±1.70● | 94.86±1.58● | 95.01±2.90 |
| breast-cancer | 96.09±0.84● | 96.57±0.68● | 96.29±0.84● | 96.53±0.79● | 96.46±1.00● |
| australian | 84.30±1.74● | 86.09±2.34● | 84.14±1.87● | 83.51±1.92● | 85.60±0.80● |
| diabetes | 75.68±2.12● | 72.41±2.55● | 75.60±2.15● | 74.67±2.07● | 72.52±1.72● |
| german.numer | 75.46±1.81 | 75.20±2.74● | 75.61±1.54 | 75.03±1.56● | 75.82±1.59● |
| splice | 83.94±1.82 | 81.74±2.50● | 85.03±1.83● | 84.06±1.79● | 87.26±2.44 |
| a8a | 81.76±0.73● | 82.05±0.72● | 81.82±0.69● | 82.12±0.49● | N/A |
| w7a | 97.94±0.33● | 97.84±0.38● | 98.11±0.23 | 98.14±0.31 | N/A |
| Top1 Time | 0 | 0 | 0 | 0 | 0 |
| DARFMMKL: w/t/l | 10/2/0 | 12/0/0 | 10/2/0 | 11/1/0 | 8/2/0 |

| Datasets | SimpleMKL | GRMKL | RMKL | MEMKL | DARFMMKL |
|---|---|---|---|---|---|
| mux6-ver1 | 87.37±2.40● | 88.35±2.13● | 84.22±2.13● | 84.71±1.94● | **95.07±3.48** |
| sonar | 82.58±0.00● | 85.14±3.24● | 81.71±4.61● | 83.29±5.13● | **85.90±4.63** |
| heart | 81.53±3.31● | 83.56±3.58 | 81.89±3.71● | 82.78±5.74● | **86.67±2.21** |
| semeion | 93.44±2.08● | 94.31±1.77● | 93.88±2.01● | 94.09±1.98● | **97.93±1.38** |
| ionosphere | 91.33±1.10● | 94.87±1.80● | 94.36±1.77● | 94.97±0.90 | **95.07±1.55** |
| breast-cancer | 96.52±0.92● | 96.75±0.40● | 96.75±0.12● | 97.08±0.51 | **97.15±0.48** |
| australian | 85.78±0.85● | 85.61±1.17● | 86.22±2.25 | 85.91±1.73● | **86.43±1.56** |
| diabetes | 75.88±1.97● | 76.88±1.93● | 77.15±1.52 | 76.80±2.36● | **77.20±1.62** |
| german.numer | 72.65±1.21● | 75.89±2.05 | 75.78±1.49 | 75.53±1.85● | **76.58±2.38** |
| splice | N/A | 84.83±2.01● | 82.97±1.77● | 84.20±1.53● | **87.32±1.81** |
| a8a | N/A | N/A | N/A | N/A | **84.43±0.72** |
| w7a | N/A | N/A | N/A | N/A | **98.57±0.00** |
| Top1 Time | 0 | 0 | 0 | 0 | 11 |
| DARFMMKL: w/t/l | 9/0/0 | 8/2/0 | 7/3/0 | 8/2/0 | |

*Table 2.* The average test accuracies (mean±std.) of our DARFMMKL and other MKL methods. We bold the numbers of the best accuracy for each dataset. ●/○ indicates the performance of DARFMMKL is significantly better/worse than compared method (paired $t$-tests at 95 % significance level). N/A means that the method did not yield results within 24 hours even under the most expedient conditions.

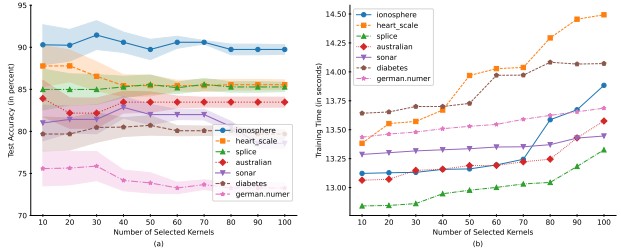

*Figure 2.* (a) Impact of $m$ on accuracy of DARFMMKL based on 7 datasets. (b) Impact of $m$ on time cost of DARFMMKL based on 7 datasets.

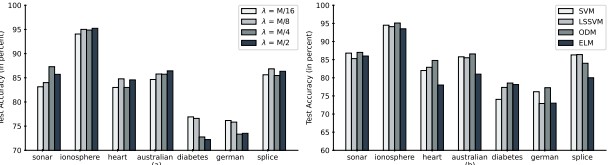

*Figure 3.* (a) Impact of $\lambda$ on accuracy of DARFMMKL based on 7 datasets. (b) Impact of classifiers on accuracy of DARFMMKL on 7 datasets.

kernel count, and very large $\lambda$ over-weights CKA at the cost of diversity, hurting accuracy.

### 6.4. The Impact of Sketch

The impact of sketch size $s$ on DARFMMKL is summarized in Figure 4 ($C = 1000$, $\lambda = M/4$, $m = 10$).

Both time cost and accuracy generally increase with $M$ and $s$: small $s$ samples too few representative instances, and larger $M$ widens the base-kernel pool. The exceptions in

Figure 4(a,c), where $M = 300$ slightly beats $M = 400$, reflect that some good kernels present at $M = 300$ are not generated at $M = 400$.

### 6.5. The Impact of Classifiers

We apply the kernels selected by DARFMMKL to four learners – SVM, LS-SVM (Suykens & Vandewalle, 1999), ELM (Huang et al., 2011), and ODM (Zhang & Zhou, 2019) – on the 7 regular-sized datasets (Figure 3(b)). All four classifiers achieve similar accuracy (ELM slightly behind), indicating that the selected kernels generalize well across kernel-based learners.

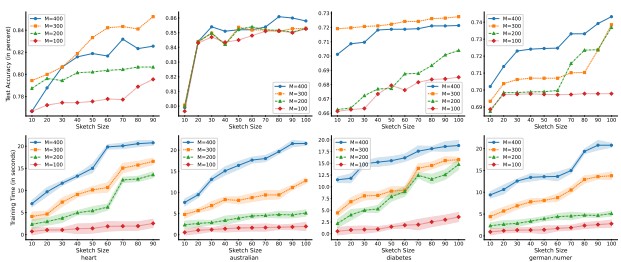

*Figure 4.* Impact of sketch size $s$ of DARFMMKL based on 4 datasets.

| Dataset | Full (q.+d.) | Quality-only | Δ |
|---|---|---|---|
| sonar | **79.05** | 78.15 | +0.90 |
| heart | **81.11** | 80.49 | +0.62 |
| breast-cancer | 96.46 | 96.46 | 0.00 |
| ionosphere | **93.16** | 91.24 | +1.92 |
| a8a | **83.96** | 83.02 | +0.94 |
| w7a | 97.84 | 97.84 | 0.00 |

*Table 4.* Quality-only vs. joint quality+diversity ablation. Accuracy (%). Removing the diversity term yields equal or worse accuracy on every dataset, with average gain +0.73 on the four datasets where the diversity term is active.

## 6.6. Ablation Study on RFM Kernels

We compare three configurations: **Traditional** (RBF + polynomial only), **RFM-Only** (RFM kernels only), and **Combined** (full DARFMMKL with both kernel types and diversity-aware selection). Results are in Table 3.

| Dataset | Traditional | RFM-Only | Combined |
|---|---|---|---|
| sonar | 82.29±3.33 | 83.71±2.10 | **85.90±4.63** |
| heart | 82.22±3.95 | 84.55±3.30 | **86.67±2.21** |
| diabetes | 75.25±2.36 | 76.31±1.63 | **77.20±1.62** |
| german | 73.07±1.52 | 74.58±1.66 | **76.58±2.38** |
| breast-cancer | 96.58±0.85 | 97.11±0.53 | **97.15±0.48** |
| ionosphere | 91.51±2.44 | 93.62±1.98 | **95.07±1.55** |
| splice | 84.34±1.90 | 85.65±1.62 | **87.32±1.81** |

*Table 3.* Ablation study on kernel types.

RFM-Only consistently beats Traditional, and Combined further improves over RFM-Only on every dataset, confirming that both AGOP-driven feature learning and diversity-aware selection contribute additively.

## 6.7. Diversity Ablation: Quality-Only vs Quality+Diversity

To isolate the contribution of the diversity term $\text{Div}(\mathcal{K})$, we compare the full objective in Eqn. (1) against a quality-only variant ($\lambda \to \infty$, top-$m$ kernels by CKA alone), holding all other settings fixed (Table 4).

The diversity term matches or improves the quality-only baseline on every dataset (avg. +0.73 on the four datasets where it is active), confirming that the gain is additive on top of the quality criterion rather than absorbed into it.

## 6.8. Additional Studies

Further analyses confirm that DARFMMKL is robust and broadly applicable: accuracy is insensitive to the RFM iteration count $T$ (Appendix C); no single Nyström landmark-sampling strategy dominates (Appendix D); on large-scale LIBSVM datasets ($n$ up to $581,012$) where dense-kernel baselines time out, DARFMMKL still outperforms sketch-adapted competitors (Appendix E); the RFM kernel family transfers to regression with consistent RMSE gains (Appendix F); on synthetic data the method recovers the four ground-truth features ($4.0/4$ on avg.) and the exact generating kernel within the top-10 candidates (Appendix G); and the BQP/LP selector is agnostic to the kernel family, accepting any PSD kernel (Appendix H).

## 7. Conclusion

We propose DARFMMKL: a data-driven RFM kernel family whose feature-importance matrix is learned via AGOP, combined with a kernel selector that jointly optimizes CKA-based quality and pairwise diversity. The resulting NP-hard BQP is reformulated as an LP and accelerated via Nyström sketching, yielding a generalization bound in which diversity tightens covering-number terms. Experiments on 12 benchmarks confirm consistent gains over 9 state-of-the-art MKL methods.

## Acknowledgements

This work was supported by the Natural Science Foundation of Wuhan (2023010201020229) and the Fundamental Research Funds for the Central Universities (No. NJ2023032).

## Impact Statement

This paper presents work whose goal is to advance the field of Machine Learning, with a particular focus on scalable and diversity-aware multiple kernel learning. By making kernel selection explicitly account for both quality and diversity, and by decoupling its computational cost from the sample size, the proposed framework can lower the barrier for applying kernel methods to large-scale tabular and scientific datasets. There are many potential societal consequences of our work, none of which we feel must be specifically highlighted here.

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

# A. Proof of Proposition 3.1

*Proof.* Throughout, we use that $\mathcal{C} = \{\mathbf{W} \in \mathbb{R}^{d \times d} : \mathbf{W} \succeq 0, \ \max_l \mathbf{W}_{ll} \leq 1\}$ is compact and convex. Convexity is immediate, since $\mathcal{C}$ is the intersection of the PSD cone with the half-spaces $\{\langle \mathbf{e}_l \mathbf{e}_l^\top, \mathbf{W} \rangle \leq 1\}$ for $l \in [d]$. For compactness, $\mathbf{W} \succeq 0$ implies $|\mathbf{W}_{ll'}| \leq \sqrt{\mathbf{W}_{ll} \mathbf{W}_{l'l'}} \leq 1$ by Cauchy–Schwarz, so $\mathcal{C}$ is bounded; and the constraints $\mathbf{W} \succeq 0$ and $\mathbf{W}_{ll} \leq 1$ are closed, so $\mathcal{C}$ is closed. Hence $\mathcal{C}$ is closed and bounded in $\mathbb{R}^{d \times d}$, and is therefore compact.

*(i) PSD of the AGOP.* For any $\mathbf{v} \in \mathbb{R}^d$ and any $i \in [n]$,

$$\mathbf{v}^\top \mathbf{J}(\mathbf{x}_i; \mathbf{W})^\top \mathbf{J}(\mathbf{x}_i; \mathbf{W}) \mathbf{v} = \|\mathbf{J}(\mathbf{x}_i; \mathbf{W}) \mathbf{v}\|_2^2 \geq 0,$$

so each $\mathbf{J}(\mathbf{x}_i; \mathbf{W})^\top \mathbf{J}(\mathbf{x}_i; \mathbf{W})$ is PSD. Therefore $\mathbf{P}(\mathbf{W}) = \frac{1}{n} \sum_{i=1}^n \mathbf{J}(\mathbf{x}_i; \mathbf{W})^\top \mathbf{J}(\mathbf{x}_i; \mathbf{W})$ is a non-negative average of PSD matrices and hence is itself PSD.

*(ii) Self-map property.* The denominator $\max_l \mathbf{P}_{ll}(\mathbf{W}) + \epsilon$ in $\Phi(\mathbf{W}) = \mathbf{P}(\mathbf{W})/(\max_l \mathbf{P}_{ll}(\mathbf{W}) + \epsilon)$ is strictly positive for any $\epsilon > 0$, so $\Phi(\mathbf{W})$ is well-defined. Dividing a PSD matrix by a positive scalar yields a PSD matrix, so $\Phi(\mathbf{W}) \succeq 0$. Moreover, by definition,

$$\max_l \Phi(\mathbf{W})_{ll} = \frac{\max_l \mathbf{P}_{ll}(\mathbf{W})}{\max_l \mathbf{P}_{ll}(\mathbf{W}) + \epsilon} < 1.$$

Hence $\Phi(\mathbf{W}) \in \mathcal{C}$ for all $\mathbf{W} \in \mathcal{C}$, i.e., $\Phi(\mathcal{C}) \subseteq \mathcal{C}$.

*(iii) Continuity.* The map $\Phi$ factors as the chain

$$\mathbf{W} \mapsto \mathbf{K_W} \mapsto (\mathbf{K_W} + \lambda_{\mathrm{rfm}} \mathbf{I})^{-1} \mapsto \boldsymbol{\alpha}(\mathbf{W}) \mapsto \mathbf{J}(\mathbf{W}) \mapsto \mathbf{P}(\mathbf{W}) \mapsto \Phi(\mathbf{W}),$$

and we verify continuity link by link. Each entry $[\mathbf{K_W}]_{ij} = \kappa_{\mathbf{W}, \sigma(\mathbf{W})}(\mathbf{x}_i, \mathbf{x}_j)$ is a continuous function of $\mathbf{W}$ since $\kappa$ is continuous in its arguments, $\sigma(\mathbf{W})$ is continuous by (A2), and the data are bounded by (A1). Matrix inversion is continuous on the open set of invertible matrices, and $\mathbf{K_W} + \lambda_{\mathrm{rfm}} \mathbf{I}$ has minimum eigenvalue at least $\lambda_{\mathrm{rfm}} > 0$ on $\mathcal{C}$ by (A3); therefore $\mathbf{W} \mapsto (\mathbf{K_W} + \lambda_{\mathrm{rfm}} \mathbf{I})^{-1}$ is continuous, and the KRR coefficients $\boldsymbol{\alpha}(\mathbf{W})$ are continuous as a composition with a fixed label vector. By (A4), differentiating the kernel function with respect to its inputs gives the entries of $\mathbf{J}(\mathbf{x}; \mathbf{W})$ as continuous functions of $\mathbf{W}$, so $\mathbf{P}(\mathbf{W})$ is continuous as a finite average. Finally, $\max_l \mathbf{P}_{ll}(\mathbf{W}) + \epsilon \geq \epsilon > 0$ on $\mathcal{C}$, so the rescaling step is continuous and $\Phi$ is continuous on $\mathcal{C}$.

*(iv) Brouwer fixed-point theorem.* $\mathcal{C}$ is compact and convex, and $\Phi : \mathcal{C} \to \mathcal{C}$ is continuous by (i)–(iii). Brouwer's fixed-point theorem (Brouwer, 1911) therefore guarantees the existence of $\mathbf{W}^\star \in \mathcal{C}$ such that $\Phi(\mathbf{W}^\star) = \mathbf{W}^\star$. $\square$

# B. Component-Wise Error Decomposition

Theorem 5.4 characterizes the kernel-selection step, but the full DARFMMKL pipeline contains two additional moving pieces – the AGOP-based RFM kernel construction and the Nyström sketch – whose roles in the prediction error are conceptually distinct. We do not claim a single end-to-end excess-risk theorem that simultaneously isolates all three effects; rather, we make explicit *which* component each part of the analysis controls.

**(i) Kernel selection.** The diversity-aware selector reduces the candidate family $\mathcal{M}$ to a subset $\mathcal{K} \subseteq \mathcal{M}$ of size $m$ whose kernels are individually predictive and mutually less redundant. The covering-number argument behind Theorem 5.4 shows that, for fixed quality, larger effective diversity $q$ can reduce the denominator-dominated covering-number terms, so the selected subset improves the approximation–complexity tradeoff relative to using the redundant pool. This is the only component for which the present paper establishes a closed-form generalization bound.

**(ii) RFM feature learning.** The learned AGOP matrix $\mathbf{W}$ induces a task-adaptive Mahalanobis geometry, which changes the effective kernel class. When $\mathbf{W}$ is better aligned with the discriminative directions of the target, the induced kernels achieve a lower approximation bias than a feature-agnostic kernel family. We support this component algorithmically (the fixed-point existence in Proposition 3.1) and empirically (the synthetic recovery study in Appendix G and the ablation in Table 3), rather than via a separate oracle-style excess-risk theorem for $\mathbf{W}$.

**(iii) Nyström sketching.** Sketching introduces a controlled kernel perturbation $\mathbf{K} \to \tilde{\mathbf{K}}$, which propagates into the trained predictor and ultimately into the risk. The standard Nyström approximation bound (Drineas & Mahoney, 2005)

$$\|\mathbf{K} - \tilde{\mathbf{K}}\|_F \leq \|\mathbf{K} - \mathbf{K}_k\|_F + O\left(\sqrt{n/s}\,\|\mathbf{K} - \mathbf{K}_k\|_2\right),$$

where $\mathbf{K}_k$ is the best rank-$k$ approximation, governs this perturbation. Because the selector relies on the *relative* ranking of kernels by quality and diversity rather than on exact values, moderate sketch sizes are sufficient in practice even when the absolute approximation error is non-negligible.

In short, Theorem 5.4 pins down the kernel-selection contribution; Proposition 3.1 together with the empirical evidence in Appendix G pins down the RFM step; and the Nyström bound above pins down the sketching step. A unified end-to-end excess-risk analysis that combines all three under a single set of assumptions is left to future work.

## C. Sensitivity to RFM Iteration Count $T$

The RFM update is iterated for a fixed number of rounds $T$ before the AGOP matrix $\mathbf{W}_T$ is plugged into the kernel pool. We first report the empirical convergence behavior in Frobenius norm (Table 5), then sweep $T \in \{1, 2, 3, 5, 10\}$ on four representative datasets while holding all other hyperparameters fixed (Table 6).

| $T$ | splice | ionosphere |
|---|---|---|
| 1 | 11.4419 | 9.1489 |
| 2 | $1.88 \times 10^{-4}$ | $2.97 \times 10^{-2}$ |
| 3 | $5.33 \times 10^{-6}$ | $1.55 \times 10^{-5}$ |
| 5 | $1.42 \times 10^{-5}$ | $1.06 \times 10^{-5}$ |

*Table 5.* Frobenius norm of successive iterates $\|\mathbf{W}_T - \mathbf{W}_{T-1}\|_F$. The learned feature matrix stabilizes within a handful of iterations.

| Dataset | $T{=}1$ | $T{=}2$ | $T{=}3$ | $T{=}5$ | $T{=}10$ |
|---|---|---|---|---|---|
| sonar | 80.00 | 79.52 | 79.05 | 78.10 | 80.00 |
| heart | 79.36 | 81.48 | **82.22** | 81.48 | 81.11 |
| breast-cancer | 94.52 | **96.64** | 96.49 | 96.51 | 96.64 |
| ionosphere | 91.98 | 92.02 | 93.16 | **94.30** | 92.31 |

*Table 6.* Test accuracy (%) for different RFM iteration counts $T$.

Combined with the rapid Frobenius-norm convergence in Table 5, this sensitivity sweep shows that small values such as $T = 3$ already provide a good operating point in practice; aggressive increases in $T$ neither markedly help nor hurt accuracy, and merely add training overhead.

## D. Nyström Landmark-Sampling Strategies

The default sketch in this paper draws landmarks via uniform random sampling. To assess the impact of this choice, we compare three landmark-selection strategies under otherwise identical settings: *Uniform* (the default), *Stratified* (class-balanced uniform sampling), and $K$-*means* (cluster-centroid landmarks). Results on four datasets are summarized in Table 7.

No single sampling strategy dominates across all datasets: uniform sampling is best on heart and breast-cancer, stratified sampling on sonar and ionosphere, while $K$-means is competitive but never strictly best in this set. The performance gap confirms that representative-landmark selection meaningfully affects accuracy and is worth tuning when accuracy is critical, but the framework itself is agnostic to the specific Nyström variant employed.

## E. Performance on Large-Scale Datasets

The benchmarks used in the main paper are small to medium scale (up to $\sim 50{,}000$ points), which is the regime where many MKL baselines are tractable. To test scalability into the regime where dense $n \times n$ kernel matrices are prohibitive, we

| Dataset | Uniform | Stratified | $K$-means |
|---|---|---|---|
| sonar | 74.76 | **79.05** | 78.57 |
| heart | **81.11** | 80.21 | 80.00 |
| breast-cancer | **97.22** | 96.49 | 97.08 |
| ionosphere | 89.46 | **92.01** | 88.17 |

*Table 7.* Test accuracy (%) under three Nyström landmark-sampling strategies. No single strategy dominates: representative-landmark selection meaningfully affects results in a dataset-dependent way.

additionally evaluate DARFMMKL on two large LIBSVM datasets, ijcnn1 ($n = 141{,}691$, $d = 22$) and covtype.binary ($n = 581{,}012$, $d = 54$). On these scales, the standard dense-kernel baselines used in Table 2 (SimpleMKL, GRMKL, RMKL, MEMKL) fail to complete within 24 hours; we therefore compare against sketch-adapted variants of three competitive MKL methods, namely Sketch-RBMKL, Sketch-ABMKL, and Sketch-EasyMKL, all of which use the same Nyström sketch as DARFMMKL. Table 8 reports test accuracy.

| Dataset | $n$ | $d$ | DARFMMKL | Sketch-RBMKL | Sketch-ABMKL | Sketch-EasyMKL |
|---|---|---|---|---|---|---|
| ijcnn1 | 141,691 | 22 | **97.37** | 96.68 | 96.52 | 97.06 |
| covtype.binary | 581,012 | 54 | **80.06** | 79.83 | 79.69 | 79.69 |

*Table 8.* Test accuracy (%) on large-scale LIBSVM datasets. Standard dense-kernel baselines (SimpleMKL/GRMKL/RMKL/MEMKL) all time out at 24 hours and are omitted; all four columns on the right share the same Nyström sketch budget as DARFMMKL.

Across both large-scale datasets, DARFMMKL outperforms all three sketch-adapted MKL baselines while sharing the same sketch budget. This indicates that the gains observed on small/medium datasets do carry over to the scale regime where sketching is essential, and that the joint quality-plus-diversity selector remains the dominant factor once the sketching pipeline is held fixed.

## F. Regression Tasks

While DARFMMKL is presented in the binary classification setting, the RFM kernel family and the diversity-aware selector are not classifier-specific. To verify this, we apply the RFM-weighted kernels of DARFMMKL inside two standard regression learners – support vector regression (SVR) and kernel ridge regression (KRR) – and compare against an RBF baseline using the same regression learner and tuning protocol. Table 9 reports RMSE on two LIBSVM regression datasets.

| Dataset | Metric | Baseline | RFM-weighted |
|---|---|---|---|
| housing | RMSE (SVR) | 4.632 | **4.313** |
| housing | RMSE (KRR) | 4.029 | **3.707** |
| cpusmall | RMSE (SVR) | 5.168 | **3.369** |
| cpusmall | RMSE (KRR) | 3.384 | **3.110** |

*Table 9.* Regression RMSE (lower is better). RFM-weighted kernels reduce error against the RBF baseline across both datasets and both regressors.

The RFM-weighted variant consistently reduces RMSE, with the largest gain on cpusmall under SVR ($5.168 \rightarrow 3.369$). This supports the broader claim that the RFM kernel family is applicable beyond classification; a full MKL regression benchmark is left as a natural future extension.

## G. Synthetic Feature and Kernel Recovery

The empirical results in the main paper are predominantly accuracy-oriented and do not directly test whether DARFMMKL recovers the *underlying* discriminative structure – the relevant features and the generating kernel. Two controlled synthetic studies are conducted to address this.

**Feature recovery.** A dataset is generated with $n = 2000$ samples and $d = 20$ features, in which only the first four features are informative. Labels are produced from the score $1.8x_0 - 1.5x_1 + 1.2\,x_2x_3 + \xi$ (two linear main effects, one pairwise interaction, plus 16 noise dimensions) and a stratified $70/30$ split. Across 5 random seeds, the top-4 feature indices ranked by $\mathrm{diag}(\mathbf{W})$ from the learned RFM/AGOP metric recover $4.0/4$ *on average*, i.e. exactly the four ground-truth informative features. Downstream accuracy with an RBF SVM also rises markedly from $90.47\%$ (baseline RBF) to $95.20\%$ (RFM-weighted RBF), confirming that the recovered metric is informative beyond a binary feature selection signal.

**Kernel recovery.** A complementary experiment tests whether DARFMMKL identifies the *generating* kernel from a substantially larger heterogeneous candidate pool. We synthesize binary classification data with $n = 1800$ and $d = 20$, where the labels are obtained by thresholding a known-kernel function evaluated against 8 random support points (with additive Gaussian noise of standard deviation 0.01). The candidate pool contains 50 kernels in total: 13 RBF kernels with $\gamma \in [0.01, 13.025]$, 13 Laplace kernels with the same $\gamma$ range, 12 polynomial kernels of degree 2 to 13, and 12 sigmoid kernels with $\gamma \in [10^{-3}, 10]$. We repeat over 10 random seeds and report how often the exact true generating kernel appears in the top-ranked candidate set.

| True kernel | Top-5 | Top-10 | Top-15 |
|---|---|---|---|
| RBF, $\gamma$=0.05 | 90% | 90% | 100% |
| Laplace, $\gamma$=0.05 | 100% | 100% | 100% |
| Polynomial, deg. 2 | 100% | 100% | 100% |
| Sigmoid, $\gamma$=0.001 | 100% | 100% | 100% |

*Table 10.* Frequency with which the exact generating kernel appears in the top-ranked candidates of DARFMMKL, over 10 seeds. The selector retrieves the true kernel inside the top-10 on every trial except for the most ambiguous RBF-$\gamma$=0.05 case, where the top-15 window is sufficient.

Since DARFMMKL is designed to choose a kernel *subset* rather than a single best kernel, recovery in a small top-ranked set is the appropriate criterion. Under this criterion the selector identifies the true generating kernel with very high frequency across all four kernel families. Together, the feature-recovery and kernel-recovery experiments provide direct, controlled evidence that the components of DARFMMKL track the latent structure of the problem rather than merely fitting noise.

## H. Beyond Gaussian and Laplacian Kernels

Although the RFM kernel family in Section 3.1 is instantiated with Gaussian and Laplacian forms, the broader DARFMMKL selector is *not* restricted to translation-invariant kernels. The candidate pool $\mathcal{M}$ already mixes RBF, polynomial, and RFM kernels in the experiments above, and any additional positive semi-definite kernel can be plugged in without changing the BQP/LP machinery – the diversity term $\mathrm{Diff}(\kappa_i, \kappa_j)$ depends only on the predictions of two kernel-induced classifiers, and the CKA-based quality term depends only on the kernel matrices, both of which are well-defined for arbitrary PSD kernels. The reason the RFM construction itself is most naturally instantiated with translation-invariant kernels is that the learned AGOP matrix $\mathbf{W}$ defines a Mahalanobis distance, which slots cleanly into Gaussian and Laplacian forms. Note that the Gaussian kernel can be viewed as an infinite-order polynomial kernel; in this sense, the Gaussian-RFM kernel implicitly captures nonlinear feature interactions of all polynomial orders while learning their relative importance through $\mathbf{W}$, which mitigates the apparent restriction.

