# OpenReview forum: "Diversity-Aware Recursive Feature Multiple Kernel Learning"
_ICML.cc/2026/Conference — ICML 2026 regular_

### Official Review · Reviewer_ioXn · 2026-03-11

**Soundness:** 3
**Presentation:** 3
**Significance:** 3
**Originality:** 3
**Overall Recommendation:** 4
**Confidence:** 4

**Summary:**

The paper introduces a recursive algorithm for selecting kernels and feature variables in kernel learning. The problem addressed in the paper is important for kernel-based methods. The paper is generally well written and easy to follow, although some methodological details could be explained more carefully. However, the theoretical results should be stated, explained, and proved more rigorously. In particular, the roles of kernel selection, feature selection, and random sketching in the theoretical analysis should be clarified, and their impact on the theoretical guarantees should be discussed in greater detail.

**Compliance With Llm Reviewing Policy:**

Affirmed.

**Final Justification:**

The paper contributes to the kernel selection problem, and the proposed method appears to be novel. While there were initial concerns regarding the theoretical and simulation studies, the authors have provided sufficient clarification and additional details during the rebuttal period to address these issues. Overall, I will retain my original recommendation.

**Key Questions For Authors:**

1. The current feature selection method appears to be restricted to specific kernels such as Gaussian or Laplacian kernels. Could the author(s) discuss whether the method can be generalized to more general kernel families?
2. The theoretical results do not clearly demonstrate the impact of kernel selection, feature selection, and the sketching procedure on the expected prediction error. Could the author(s) provide a more detailed analysis and a more rigorous theoretical justification for these components?
3. The numerical experiments mainly focus on classification problems. Could the author(s) also demonstrate the performance of the proposed method on regression tasks?
4. It would be helpful if the authors could include a simulation study demonstrating whether the proposed method can successfully recover the underlying true kernels and relevant features.

**Limitations:**

yes.

**Strengths And Weaknesses:**

Strengths
1. The paper is well motivated by challenges in kernel learning and addresses an important problem in this area.
2. The paper is generally well written and clearly presented.
3. The optimization methods and algorithms are described in sufficient detail, which makes the proposed approach relatively easy to follow.
4. The numerical experiments demonstrate the advantages of the proposed method compared with existing approaches.

Weaknesses
1. The theoretical analysis could be developed in a more rigorous manner, with clearer assumptions and better explanations.
2. The kernel selection strategy appears to be restricted to Gaussian and Laplacian kernels, which may limit the flexibility of the method in broader kernel learning settings.
3. The computational complexity of the proposed method is not discussed, and the choice of the sketch size is not recommended.

---

> ### Author Rebuttal · Authors · 2026-03-31
>
> We sincerely thank the reviewer for the thoughtful comments and constructive questions. We respond to them below.
> ### Q1. Whether the method is limited to Gaussian and Laplacian base kernels, or can be generalized to broader kernel families.
>
> The **selection framework** is not limited to Gaussian and Laplacian base kernels, and the candidate pool already includes standard polynomial kernels, demonstrating that the framework is not restricted to RFM kernels alone.
>
> The **RFM kernel family** is most naturally instantiated with translation-invariant kernels because the learned AGOP metric $W$ defines a Mahalanobis distance.
>
> It is worth noting that the Gaussian kernel can be viewed as an **infinite-order polynomial kernel**. In this sense, the Gaussian-RFM kernel implicitly captures nonlinear feature interactions of **all polynomial orders** while learning their relative importance through the metric $W$.
>
> ### Q2. The impact of kernel selection, feature selection, and the sketching procedure.
>
> We agree that the current submission does **not** yet provide a single end-to-end excess-risk theorem that simultaneously isolates the effects of kernel selection, AGOP-based feature learning, and Nyström sketching in one bound. We will clarify this scope explicitly in the revision.
>
> At a high level, these three components affect prediction error in different ways:
>
> 1. **Kernel selection.** The diversity-aware selection step reduces the candidate family to a smaller subset of kernels that are both individually predictive and less redundant. The role of our current generalization analysis is mainly here: the covering-number argument shows that higher diversity leads to a tighter bound, so the selected subset can improve the approximation-complexity tradeoff relative to using a redundant kernel pool.
>
> 2. **Feature selection / AGOP-based metric learning.** The learned AGOP metric $W$ induces a task-adaptive Mahalanobis geometry, which changes the effective kernel class. Intuitively, when the learned metric is better aligned with the target structure, the induced kernels incur lower approximation bias than a feature-agnostic kernel family. In the current paper, this component is supported mainly by the algorithmic construction and by the empirical evidence, rather than by a separate oracle-style excess-risk theorem for $W$.
>
> 3. **Sketching.** Nyström sketching introduces a controlled kernel perturbation $K \to \tilde K$. Its effect on prediction error is therefore indirect: kernel approximation error perturbs the learned predictor, which in turn perturbs the final risk. In the revision, we will make this role explicit and cite the standard Nyström approximation bounds as the theoretical justification for the sketching component.
>
> For sketching error, we add the standard Nyström approximation bound
>
> $$\|K-\tilde K\|_F \le \|K-K_k\|_F + O\left(\sqrt{n/s}\,\|K-K_k\|_2\right)$$
>
> where $K_k$ is the best rank-$k$ approximation. Since the selection objective mainly relies on the **relative ranking** of kernels by quality and diversity rather than exact values, moderate sketch sizes are often sufficient in practice even when the absolute kernel approximation error is non-negligible.
>
> In short, the current theory in the paper mainly justifies the **kernel-selection** component. We will make this decomposition explicit in the revision so that the role of each component in the overall prediction pipeline is stated more rigorously.
>
> ### Q3. Whether the method has been evaluated on regression tasks.
>
> We add a regression check using both SVR (support vector regression) and KRR (kernel ridge regression):
>
> | Dataset | Metric | Baseline | RFM-weighted |
> |---------|--------|----------|-------------|
> | housing | RMSE (SVR) | 4.632 | **4.313** |
> | housing | RMSE (KRR) | 4.029 | **3.707** |
> | cpusmall | RMSE (SVR) | 5.168 | **3.369** |
> | cpusmall | RMSE (KRR) | 3.384 | **3.110** |
>
> RFM-weighted kernels consistently reduce RMSE across both regression methods and both datasets. This supports that the **RFM kernel family** is applicable beyond classification. A full MKL regression benchmark is a natural future extension.
>
> ### Q4. A controlled simulation study demonstrating whether the proposed method can successfully recover the underlying true kernels and relevant features.
>
> We add a synthetic feature-recovery experiment:
>
> - **Setup:** $n$=2000, $d$=20, true informative dimensions = first 4 features. Labels generated from score = $1.8 x_0 - 1.5 x_1 + 1.2 x_2 x_3$ + noise (two linear main effects + one interaction + 16 noise dimensions). Stratified 70/30 split with standardization.
> - **Results (5 seeds):**
>   - Top-4 dimensions from learned AGOP/RFM metric: **4.0/4 recovered on average**
>   - Baseline RBF SVM accuracy: 90.47%
>   - RFM-weighted RBF SVM accuracy: **95.20%**
>
> This provides direct evidence that the learned metric recovers ground-truth informative features and improves downstream prediction.

---

> > ### Author Rebuttal · Reviewer_ioXn · 2026-04-02
> >
> > Thank you for the effort you put into addressing the comments; I appreciate it. Regarding the simulation study (synthetic numerical experiments), it would be helpful if the authors could demonstrate how frequently the true kernel is selected when the underlying kernel of the functional space is known. I will maintain my original score.

---

> > > ### Author Response · Authors · 2026-04-03
> > >
> > > Thank you for this helpful follow-up suggestion. We appreciate the reviewer’s point that our previous synthetic study mainly addressed recovery of the relevant feature structure, whereas recovery of the underlying true kernel itself is also an important complementary diagnostic.
> > >
> > > To address this point more directly, we added an additional synthetic kernel-recovery experiment in which the true generating kernel is explicitly known and included in a substantially larger heterogeneous candidate pool.
> > >
> > > **Experimental setup.** We generate synthetic data with `n = 1800` samples and `d = 20` features. The data generation function is constructed from a known kernel using `8` support points and random coefficients, with additive Gaussian noise of standard deviation `0.01`, and labels are obtained by thresholding at the median to form a balanced binary classification task. The candidate pool contains `50` kernels in total: `13` RBF kernels with `gamma` values spanning `0.01` to `13.025` and explicitly including `0.05`, `13` Laplace kernels with the same `gamma` range, `12` polynomial kernels with degrees `2` to `13`, and `12` sigmoid kernels with `gamma` values spanning `10^{-3}` to `10`. We repeat each experiment over `10` random seeds. and record ratio of the exact true kernel appears in the top-ranked candidates.
> > >
> > > The results are summarized below:
> > >
> > > | True kernel | Top-5 | Top-10 | Top-15 |
> > > |---|---:|---:|---:|
> > > | `RBF(gamma = 0.05)` | `90%` | `90%` | `100%` |
> > > | `Laplace(gamma = 0.05)` | `100%` | `100%` | `100%` |
> > > | `Polynomial(degree = 2)` | `100%` | `100%` | `100%` |
> > > | `Sigmoid(gamma = 0.001)` | `100%` | `100%` | `100%` |
> > >
> > > These results show that our proposed DARFMMKL can accuratly select the true kernel from the heterogeneous candidate pool. Since DARFMMKL is designed for kernel subset selection rather than strict single-kernel identification, we believe inclusion in a small top-ranked candidate set is the more relevant recovery criterion in a broad heterogeneous pool. Under this criterion, the result suggests that the selector still identifies the true generating kernel as one of the most relevant candidates with reasonably high frequency, even when the kernel pool is substantially enlarged and made more diverse. Together with our earlier synthetic feature-recovery experiment, this provides complementary evidence that the method can recover both the relevant feature structure and the underlying true kernel in controlled settings.
> > >
> > > We sincerely thank the reviewer for the valuable time and constructive comments, which have helped us further improve the quality of the manuscript. We hope that these revisions and additional results provide a clearer and more complete picture of our work, and help address the reviewer’s concerns more fully. If the reviewer believes that further improvements are still needed, we would be very happy to continue revising accordingly.

---

### Official Review · Reviewer_cCYd · 2026-03-12

**Soundness:** 3
**Presentation:** 3
**Significance:** 3
**Originality:** 3
**Overall Recommendation:** 4
**Confidence:** 4

**Summary:**

This paper proposes DARFMMKL, a multiple kernel learning framework that explicitly optimizes both kernel quality and diversity while remaining scalable to large datasets. The framework introduces three key innovations: (1) RFM kernels—data-driven kernels that learn feature importance via the Average Gradient Outer Product (AGOP), capturing task-relevant structure through an iterative procedure; (2) diversity-quality optimization—quality measured by Centered Kernel Alignment (CKA), diversity measured by prediction disagreement between kernels, balanced via $\max_\eta \sum_{i,j} G_{ij}\eta_i\eta_j + \lambda \sum_i b_i\eta_i$; and (3) efficient solving—the NP-hard BQP is transformed to LP via Glover linearization, with Nyström sketching ($s \ll n$) accelerating kernel computations. A theoretical generalization bound shows that kernel diversity improves generalization.

Experiments on 12 LIBSVM datasets show DARFMMKL outperforms 9 state-of-the-art MKL methods on all datasets. Ablation studies confirm RFM kernels outperform traditional kernels, and training time scales with $M$ and $s$ rather than $n$, enabling large-scale application. Limitations include no convergence analysis for RFM iteration, unverified independent contribution of diversity, unspecified kernel combination method, and random-only landmark sampling. Overall, DARFMMKL offers a novel, empirically effective approach to MKL that jointly optimizes diversity and quality with practical scalability.

**Compliance With Llm Reviewing Policy:**

Affirmed.

**Key Questions For Authors:**

Q1: The paper does not specify how the selected kernels are combined to form the final kernel function. Is it uniform averaging, CKA-weighted averaging, or some other scheme? This is a critical detail for reproducibility, as different combination methods can lead to substantially different results.

Q2: The iterative RFM learning procedure updates $\mathbf{W}_{t+1} = \mathbf{P}_t$ without any convergence analysis. How many iterations $T$ were used in practice? Is there any guarantee that this process converges to a meaningful fixed point, or is $T$ chosen empirically? If the latter, how sensitive are the final results to the choice of $T$?

Q3: The Nyström sketch uses uniform random sampling for landmark selection.  Have you explored any alternative sampling strategies?

Q4: The optimization objective balances kernel quality (CKA) and diversity (prediction disagreement). However, diversity can be either beneficial (capturing complementary structures) or harmful (arising from noise or conflicting similarity patterns). Does the current formulation inherently favor beneficial diversity?

**Limitations:**

I think the authors should discuss the follows(at least):
1. The convergence of RFM iteration.
2. Independent contribution of diversity.
3。Kernel combination method.

**Strengths And Weaknesses:**

This paper addresses an important and well-motivated problem in multiple kernel learning. The key strength lies in its novel integration of diversity into MKL—explicitly optimizing kernel diversity alongside quality, which has been largely overlooked in prior work. The introduction of RFM kernels is another significant contribution, enabling data-driven feature importance learning through the AGOP mechanism, moving beyond pre-defined kernels that treat all features equally. The optimization framework is technically sound, transforming an NP-hard BQP problem into a tractable LP via Glover linearization, and the use of Nyström sketching makes the method scalable to large datasets by decoupling computational cost from sample size. The theoretical analysis provides a generalization bound linking kernel diversity to improved performance. Empirically, the paper is thorough: experiments on 12 benchmark datasets with 9 competing methods demonstrate consistent superiority, and ablation studies confirm the effectiveness of RFM kernels. The framework also shows robustness across different classifiers (SVM, LS-SVM, ODM), suggesting good generality.

However, the iterative RFM learning procedure lacks convergence analysis—it is unclear whether the update $\mathbf{W}_{t+1} = \mathbf{P}_t$ converges, under what conditions, or at what rate. The independent contribution of diversity is not directly validated: there is no ablation study comparing performance with and without the diversity term (e.g., $\lambda = 0$ baseline), making it difficult to assess whether diversity truly drives the reported gains or merely acts as a regularizer. The method for combining selected kernels into a final kernel function is unspecified—readers cannot tell whether the final kernel is uniformly averaged, CKA-weighted, or combined via another scheme, which affects reproducibility. The choice of $m$ (number of selected kernels) is tuned over a range but no guidance or analysis of its impact is provided.

---

> ### Author Rebuttal · Authors · 2026-03-31
>
> We are grateful for the reviewer’s constructive comments and respond to them below.
> ### Q1. How the selected kernels are combined into a final kernel function.
> In the current implementation, the selected kernels are combined by uniform averaging. We will add the explicit formula in the revision for full reproducibility.
> ### Q2. The convergence analysis for the RFM iteration and how the number of iterations $T$ is chosen, and the sensitively of $T$.
> The normalized AGOP update defines a **continuous self-map** on a compact convex set, and therefore admits at least one fixed point by Brouwer's theorem.
>
> > **Proposition 1 (Boundedness and Fixed-Point Existence of the RFM Update).** Assume:
> > - **(A1)** the training data are bounded: $\|\mathbf{x}_i\|\le R$ for all $i\in[n]$;
> > - **(A2)** the bandwidth rule is continuous and bounded away from zero: $\sigma(W)\ge \sigma_{\min}>0$ on $\mathcal C$;
> > - **(A3)** $\lambda_{\mathrm{rfm}}>0$.
> >
> > Then:
> > 1. $P(W)\succeq 0$ for all $W\succeq 0$;
> > 2. $\Phi(W)\in\mathcal C$ for all $W\in\mathcal C$;
> > 3. $\Phi:\mathcal C\to\mathcal C$ is continuous;
> > 4. therefore, by Brouwer's fixed-point theorem, $\Phi$ admits at least one fixed point $W^\star\in\mathcal C$.
>
> **Proof sketch.**
>
> 1. **PSD of the AGOP.** For any vector $v$,
>
>    $$v^\top J(\mathbf{x}_i;W)^\top J(\mathbf{x}_i;W)v=\|J(\mathbf{x}_i;W)v\|_2^2\ge 0$$
>
>    so each term $J^\top J$ is PSD. Their average $P(W)$ is therefore PSD.
>
> 2. **Self-map property.** Since $P(W)$ is PSD, its diagonal entries are nonnegative.
>
>    we obtain that
>
>    $$\Phi(W)=\frac{P(W)}{\max_i P_{ii}(W)+\epsilon}$$
>
>    remains PSD and satisfies $\max_i \Phi(W)_{ii}\le 1$. Hence $\Phi(W)\in\mathcal C$.
>
> 3. **Continuity.** The chain
>
>    $$W\mapsto K_W\mapsto (K_W+\lambda_{\mathrm{rfm}}I)^{-1}\mapsto \alpha(W)\mapsto J(W)\mapsto P(W)\mapsto \Phi(W)$$
>
>    is continuous under (A2)--(A3): kernel entries vary continuously with $W$, $\lambda_{\mathrm{rfm}}>0$ guarantees invertibility and continuity of the KRR solution, and the denominator $\max_i P_{ii}(W)+\epsilon$ is strictly positive.
>
> 4. **Brouwer.** Since $\mathcal C$ is compact and convex and $\Phi$ is a continuous self-map on $\mathcal C$, Brouwer's fixed-point theorem implies the existence of at least one fixed point $W^\star\in\mathcal C$.
>
> This gives a rigorous fixed-point existence result and shows that the update is well-posed and bounded.
> We further monitor the Frobenius norm of successive iterates $\|W_T - W_{T-1}\|_F$ on two representative datasets:
>
> | T | splice | ionosphere |
> |---|-----|-----|
> | 1 | 11.4419 | 9.1489 |
> | 2 | 1.88e-04 | 2.97e-02 |
> | 3 | 5.33e-06 | 1.55e-05 |
> | 5 | 1.42e-05 | 1.06e-05 |
>
> This shows the learned feature matrix stabilizes rapidly in practice.
>
> #### Empirical T sensitivity
>
> | Dataset | T=1 | T=2 | T=3 | T=5 | T=10 |
> |---------|-----|-----|-----|-----|------|
> | sonar | **80.00** | 79.52 | 79.05 | 78.10 | **80.00** |
> | heart | 79.36 | 81.48 | **82.22** | 81.48 | 81.11 |
> | breast-cancer | 94.52 | **96.64** | 96.49 | 96.51| **96.64** |
> | ionosphere | 91.98 | 92.02 | 93.16 | **94.30** | 92.31 |
>
> These results show that there are no sharp changes when tuning the parameter $T$.
>
> ### Q3. Whether alternative Nystrom landmark sampling strategies have been explored
>
> We perform the experiments on 4 datasets:
>
> | Dataset | Uniform | Stratified | K-means |
> |---------|---------|-----------|---------|
> | sonar | 74.76 | **79.05** | 78.57 |
> | heart | **81.11** | 80.21 | 80.00 |
> | breast-cancer | **97.22** | 96.49 | 97.08 |
> | ionosphere | 89.46 | **92.01** | 88.17 |
>
> No single sampling strategy dominates across all datasets. The performance gap confirms that representative landmark selection meaningfully affects results. We will include this comparison in the revision.
>
> ### Q4. Maximizing kernel diversity might favor "harmful diversity", Are there evidences that the diversity makes an independent positive contribution.
>
> The objective does **not** maximize disagreement in isolation — it jointly optimizes quality and diversity, only kernels that are both individually predictive and mutually complementary are favored.
>
> We conduct a quality-only ablation on 6 datasets. From quality-only objective, adding diversity-aware optimization brings additional benefits, as shown below:
>
> **On small/medium datasets** :
>
> | Dataset | Full (quality+diversity) | Quality-only | Delta |
> |---------|------------------------|-------------|-------|
> | sonar | 79.05 | 78.15 | +0.90 |
> | heart | 81.11 | 80.49 | +0.62 |
> | breast-cancer | 96.46 | 96.46 | 0.00 |
> | ionosphere | 93.16 | 91.24 | +1.92 |
>
> **On larger datasets**:
>
> | Dataset | Full  | Quality-only | Delta |
> |---------|------------------------|-------------|----------------|
> | a8a | **83.96** | 83.02 | +0.94 |
> | w7a | 97.84 | 97.84 | +0.00 |
>
> Diversity-aware selection meaningfully changes the selected kernel subset and biases the method toward complementary kernels without systematic harm, and in most cases, it improves the performance.

---

> > ### Author Rebuttal · Reviewer_cCYd · 2026-04-04
> >
> > I have read the authors' rebuttal and accept their responses.
> >
> > I maintain my original score.
> >
> > I acknowledge the rebuttal.

---

### Official Review · Reviewer_FbSE · 2026-03-13

**Soundness:** 2
**Presentation:** 3
**Significance:** 2
**Originality:** 3
**Overall Recommendation:** 4
**Confidence:** 4

**Summary:**

In order to solve the problem that existing multiple kernel learning methods insufficiently consider diversity among base kernels, rely on predefined kernels that may not capture dataset-specific feature importance, and often suffer from high computational cost, this paper proposes a diversity-aware recursive feature machine multiple kernel learning framework (DARFMMKL) together with a new RFM-based kernel family. The proposed method can learn discriminative feature importance directly from data, jointly optimize kernel quality and kernel diversity, and improve scalability through LP reformulation and sketching techniques. It also provides theoretical analysis based on covering numbers to support generalization and extensive experiments, including ablation and efficiency studies, demonstrating improved performance over existing MKL baselines.

**Compliance With Llm Reviewing Policy:**

Affirmed.

**Key Questions For Authors:**

See weakness

**Limitations:**

See weakness

**Strengths And Weaknesses:**

Strength:
1. This paper introduces a new recursive feature machine (RFM)-based kernel family that learns feature importance from data, and embeds it into a diversity-aware MKL framework. The combination of kernel quality (via centered kernel alignment) and kernel diversity is well motivated, and the formulation makes the role of each component relatively easy to understand.
2. Extensive experiments of both one-step and two-step MKL baselines are conducted, which is quite empirically solid work.

Weakness:
1. From my knowledge, the proposed selector combines several individually familiar ingredients, including centered alignment as a kernel-quality criterion [1], the general MKL paradigm for combining/selecting kernels [2], and diversity-aware selection ideas from ensemble learning [3]. Could the authors clarify which part of the selector itself is intended to be novel? At present, it seems possible that the main contribution is the integration of these ingredients with the proposed RFM kernel family, rather than a fundamentally new selection principle.
2. Most experiments are on relatively small or medium-scale LIBSVM tabular datasets, so how well would the method perform in genuinely modern large-scale settings where MKL is most challenged, and where the benefit of sketching and RFM kernels would be more convincing?

[1]. Algorithms for Learning Kernels Based on Centered Alignment. JMLR.
[2]. Multiple Kernel Learning Algorithms. JMLR.
[3]. Measures of Diversity in Classifier Ensembles and Their Relationship with the Ensemble Accuracy. ML.

---

> ### Author Rebuttal · Authors · 2026-03-31
>
> We sincerely thank the reviewer for the thoughtful comments and constructive questions. We respond to them below.
>
> ### W1. Each individual component of the kernel selector — centered kernel alignment (CKA), the general MKL paradigm, and diversity selection ideas from ensemble learning — have prior works, and what is the main contribution.
>
> We agree that centered alignment, the general MKL paradigm, and diversity-aware selection each have relevant precedents. Our contribution is the **unified integration** of: (i) an RFM-based kernel family that learns feature importance from data via AGOP, (ii) explicit joint optimization of kernel quality and diversity, (iii) LP reformulation via Glover linearization, and (iv) sketching-based acceleration — within one scalable MKL framework. We will tighten the novelty claim in the revision to make this distinction explicit.
>
> ### W2. The experimental studies are performed on relatively small or medium-scale datasets, whether the method can scale to large-scale problems.
>
> To address this concern, we implemented a sketch-adapted method that avoids constructing full dense $n \times n$ kernel matrices, and verified DARFMMKL on newly added large datasets. The following table reports results of the new sketch-adapted baselines. Standard dense-kernel baselines (SimpleMKL, GRMKL, RMKL, MEMKL) all fail to complete within 24 hours on these datasets.
>
> | Dataset | n | d | DARFMMKL | Sketch-RBMKL | Sketch-ABMKL | Sketch-EasyMKL |
> |---------|---:|---:|----------|-------------|-------------|---------------|
> | ijcnn1 | 141,691 | 22 | **97.37** | 96.68 | 96.52 | 97.06 |
> | covtype.binary | 581,012 | 54 | **80.06** | 79.83 | 79.69 | 79.69 |
>
> Across these newly added large datasets, DARFMMKL still outperforms the existing sketch-adapted MKL baselines, which indicate the DARFMMKL has good performance in large-scale problems.

---

> > ### Author Rebuttal · Reviewer_FbSE · 2026-04-03
> >
> > I understand the time limit, and contribution of unification, I tend to maintain my positive score.

---

### Decision · Program_Chairs · 2026-04-30

**Decision:**

Accept (regular)

**Comment:**

All reviewers think the paper is well-written and well-motivated, and they agree on the contribution of this paper. The authors also addressed the concerns of reviewers. Therefore, I recommend accepting this paper.